# Synthesis, Characterization and Catalytic Property Studies for Isoprene Polymerization of Iron Complexes Bearing Unionized Pyridine-Oxime Ligands

**DOI:** 10.3390/polym14173612

**Published:** 2022-09-01

**Authors:** Mengmeng Zhao, Ying Ma, Xianhui Zhang, Liang Wang, Guangqian Zhu, Qinggang Wang

**Affiliations:** 1School of Chemistry and Chemical Engineering, Linyi University, Linyi 276000, China; 2Key Laboratory of Biobased Materials, Qingdao Institute of Biomass Energy and Bioprocess Technology, Chinese Academy of Sciences, Qingdao 266000, China; 3Key Laboratory of Structure & Materials for Aviation Tire, CHEMCHINA Shuguang Rubber Industry Research & Design Institute Co., Ltd., Guilin 541000, China

**Keywords:** iron(II) catalyst, pyridine-2-aldoxime ligand, isoprene polymerization, binary catalytic system, ternary catalytic system

## Abstract

Iron complexes of the types [Fe(HL)_2_Cl_2_] (**Fe1**: HL^1^ = pyridine-2-aldoxime; **Fe2**: HL^2^ = 6-methylpyridine-2-aldoxime; **Fe3**: HL^3^ = phenyl-2-pyridylketoxime; **Fe4**: HL^4^ = picolinaldehyde O-methyl oxime) were prepared and characterized by elemental analysis and IR spectroscopy. The crystal structure of **Fe2**, determined by single-crystal X-ray diffraction, featured a distorted octahedral coordination of the iron center binding with two ligands of HL^2^. The X-ray structure and infrared spectral data indicated that pyridine-oxime ligands act as unionized bidentate ligand by coordinating with N_pyridine_ and N_oxime_. The catalytic performance for isoprene polymerization, catalyzed by these pyridine-oxime-ligated iron complexes, was examined. For a binary catalytic system combined with MAO, complexes **Fe1**, **Fe3** and **Fe4** were found to be highly active (up to 6.5 × 10^6^ g/mol·h) in *cis*-1,4-*alt*-3,4 enchained polymerization, with average molecular weights in the range of 60–653 kg/mol and narrow PDI values of 1.7–3.5, even with very low amounts of MAO (Al/Fe = 5). Upon activation with [Ph_3_C][B(C_6_F_5_)_4_]/AlR_3_ for the ternary catalytic system, theses complexes showed extremely high activities, as well about 98% yield after 2 min, to afford *cis*-1,4-*alt*-3,4-polyisoprene with a molecular weight of 140–420 kg/mol.

## 1. Introduction

The catalytic polymerization of conjugated dienes is one of the most promising methods for manufacturing synthetic rubber to make up for the deficiency of natural rubber. From a mechanistic point of view, 1,3-dienes can be polymerized in different ways, such as radical [1], cationic [2], or coordination polymerization [3,4], all yielding polymers with diverse structures and properties. Notably, coordination polymerization catalyzed by a transition metal has attracted much attention since the discovery of Ziegler-Natta catalysts in the 1950s [5,6,7]. From the early-1960s onwards, lanthanide metal-catalyzed polymerization of conjugated dienes came to be the focus of fundamental research [8]. Later, in 1995, Brookhart and coworkers applied cationic Ni and Pd complexes with bulky diimine ligands as new catalysts, marking a milestone in the research of olefin polymerization catalyzed by late transition metals [9]. On the basis of the good performance in terms of strength, flexibility and easy processing, recent progress in diene polymerization catalyzed by early transition metal complexes [8,9,10], rare earth complexes [11,12,13,14], and late transition metal complexes [15,16,17], has made it possible to tailor various properties of rubbers according to the type of catalyst used.

Catalysts using late transition metals are promising for 1,3-diene polymerization. Historically, iron, as an abundant, first-row, late transition metal, has attracted much less attention than Ni, Co and Pd compounds, despite the fact that it is less toxic and more sustainable than other metals. Early research on the iron-catalyzed polymerization of conjugated diene focused on (1) the addition of some nitrogenous or phosphorous compounds to stabilize the iron ions and improve the activity and selectivity, and (2) on the catalytic mechanism of these coordinating compounds [18,19,20,21]. Various well-defined iron complexes coordinated with bidentate or tridentate ligands, e.g., pyridine, imine, pyrazole, oxazoline, and phosphine moieties, etc. (Figure 1), have been systematically explored in olefin polymerization and have exhibited high levels of activity and the ability to control the molecular weight and molecular weight distribution of the resultant polymers [22,23,24,25,26,27,28,29,30,31,32,33]. As part of our work developing iron-based catalysts for isoprene polymerization, various pyridine-imine and pyridine-amine iron complexes have been applied to the polymerization of conjugated dienes to probe the ligand effects [34,35,36,37,38,39]. Therefore, the discovery of a novel catalytic system—with diverse steric hindrance and electronic circumstances that affect the activity and selectivity of isoprene polymerization—is particularly significant. Presently, we are trying to identify suitable ligands with different structures toward the iron center.

Pyridine-oxime compounds are important organic ligands with unique coordination abilities and various coordination modes. They can be obtained from the Schiff base condensation of pyridine-2-aldehyde or ketone with hydroxylamine. Metal complexes chelating with pyridine-oxime ligands have long been attracted attention as bioinorganic model compounds and molecule-based magnetic materials [40,41]. However, the catalytic applications of these complexes have rarely been explored, especially in the field of olefin polymerization [42,43]. Here, we report the synthesis, structures, and activities of pyridine-oxime-ligated mononuclear iron complexes and investigate the novel catalytic performance in an iron-catalyzed isoprene polymerization process in which pyridine-oxime is used as a neutral ligand. Additionally, the function of the H atom in oxime is determined by replacing it with a methyl group. Furthermore, we investigated isoprene polymerization with Fe/Al binary and Fe/Al/[Ph_3_C][B(C_6_F_5_)_4_] ternary catalytic systems, respectively, to demonstrate the wide application potential of these pyridine-oxime-ligated iron complexes.

## 2. Materials & Methods

### 2.1. Materials

All experiments were carried out under argon atmosphere. FeCl_2_, **L1**, **L3**, 2-pyridinecarboxaldehyde, 6-methylpicolinaldehyde, hydroxylamine hydrochloride, methoxyamine hydrochloride and cocatalysts were purchased. Toluene was refluxed and distilled over sodium and stored over molecular sieves under nitrogen. Hexane, dichloromethane and isoprene were refluxed and distilled over calcium hydride and stored over molecular sieves under nitrogen. The molecular weights (*M_n_*) and molecular weight distributions (*M_w_*/*M_n_*) of polymers were measured by high temperature gel permeation chromatography (GPC) using a PL-GPC 220 chromatography and maintained at 150 °C, using trichlorobenzene as an eluent and polystyrene as a standard. The NMR spectra were measured on a Bruker Advance 400 spectrometer at 298 K. ^1^H NMR and ^13^C NMR spectra of polyisoprene were recorded in CDCl_3_ and trimethylsilane as an internal reference. The polyisoprene microstructure of the 1,4 and 3,4 ratio was determined from ^1^H NMR of the 1,4 =CH signals at 5.15 ppm and the 3,4 =CH_2_ signal at 4.7 ppm. The *trans*/*cis*-1,4 stereoisomer ratio was determined from ^13^C NMR of the –CH_3_ signals of *cis*-1,4 at 23.8 ppm and *trans*-1,4 at 16.3 ppm. The mass spectra for iron complexes were detected using an ACQUITYTM UPLC & Q-TOF MS Premier. Elemental analyses were performed using a Vario EL III elemental analyzer. X-ray diffraction data were obtained using a Smart 1000 diffractometer with a Mo K-alpha X-ray source (λ = 0.71073 Å) at 298 K. Attenuated total reflection-infrared (ATR-IR) spectroscopy was conducted using a Thermo Scientific Nicolet iN10.

### 2.2. Preparation of Ligands ***L2*** and ***L4***

Ligand **L2** was prepared using a procedure reported in [44] with a little change. Briefly, to a suspension of 6-methylpicolinaldehyde (2.0 g, 16.51 mmol) in MeOH (40 mL) was added hydroxylamine hydrochloride (1.15 g, 16.51 mmol) followed by Na_2_CO_3_ (0.97 g, 9.08 mmol). The mixture was stirred at room temperature for 4 h. Then, the orange suspension was filtered and the solid was dissolved in ethyl acetate (40 mL). The solution was transferred to a separatory funnel, washed in water (3 × 20 mL), and dried over anhydrous Na_2_SO_4_. The solvent was removed in a rotary evaporator and the product was dried under vacuum. The compound was characterized as follows: a yellow solid, 1.9 g, 84% yield; ^1^H NMR (400 MHz, CDCl_3_, 298 K) δ 8.28–8.25 (m, 2H), 7.64–7.55 (m, 2H), 7.15 (dd, *J* = 8.0, 2.0 Hz, 1H), 2.59 (s, 3H); ^13^C NMR (100 MHz, CDCl_3_, 298 K) δ 158.5, 151.0, 150.9, 136.8, 123.8, 118.2, 24.3.

Ligand **L4** was prepared similarly to ligand **L2**, i.e., by blending 2-pyridinecarboxaldehyde (5.30 g, 49.48 mmol) and methoxyamine hydrochloride (4.13 g, 49.48 mmol) in methanol (60 mL), followed by the addition of Na_2_CO_3_ (2.91 g, 27.24 mmol). The compound was characterized as follows: light yellow oil, 5.8 g, 86% yield; ^1^H NMR (400 MHz, CDCl_3_, 298 K) δ 8.61–8.59 (m, 1H), 8.15 (s, 1H), 7.77 (dt, *J* = 8.0, 1.2 Hz, 1H), 7.68 (td, J = 7.7, 1.8 Hz, 1H), 7.24 (ddd, *J* = 15.6, 8.0, 1.6 Hz, 1H), 4.02 (s, 3H). ^13^C NMR (100 MHz, CDCl_3_, 298 K) δ 151.5, 149.6, 149.0, 136.3, 123.9, 121.0, 62.4.

### 2.3. Preparation of Complexes ***Fe1***–***Fe4***

Ligand (**L1**–**L4**) (2.0 equiv.) and FeCl_2_ (1.0 equiv.) were added to anhydrous dichloromethane (10 mL) in a glovebox. The reaction mixture was stirred for 15 h at room temperature. Then, the solvent was concentrated to 2 mL and hexane (5 mL) was added, resulting in precipitation which was collected by filtration, washed with hexane (3 × 5 mL) and dried under vacuum to give the corresponding complex.

**dichloro[pyridine-2-aldoxime]iron(II) Fe1 [45]**: Claret-red solid, 229 mg, 78% yield. ATR-IR (cm^−1^): 3360, 2744, 1605, 1515, 1460, 1046, 885, 768. TOF-MS-ES+ (*m*/*z*): calcd. for [C_12_H_12_ClFeN_4_O_2_]^+^: 334.9998, found: 335.0007. Anal.: calcd. For C_12_H_12_Cl_2_FeN_4_O_2_: C, 38.85; H, 3.26; N, 15.10; found: C, 37.61; H, 3.26; N, 14.35.

**dichloro[6-methylpyridine-2-aldoxime]iron(II) Fe2 [45]**: Orange solid, 421 mg, 87% yield. ATR-IR (cm^−1^): 3072, 3003, 1650, 1600, 1491, 1320, 1031, 1006, 798. TOF-MS-ES+ (*m*/*z*): calcd. for [C_18_H_21_FeN_6_O_2_]^+^: 409.1075, found: 409.1614. Anal.: calcd. For C_14_H_16_Cl_2_FeN_4_O_2_: C, 42.14; H, 4.04; N, 14.04; found: C, 42.02; H, 4.01; N, 13.93.

**dichloro[phenyl-2-pyridylketoxime] iron(II) Fe3 [46]**: Dark brown solid, 352 mg, 85% yield. ATR-IR (cm^−1^): 3064, 1598, 1459, 1441, 1338, 1048, 1024, 957, 790, 747, 701. TOF-MS-ES+ (*m*/*z*): calcd. for [C_36_H_29_FeN_6_O_3_]^+^: 649.1651, found: 649.1646. Anal.: calcd. For C_24_H_20_Cl_2_FeN_4_O_2_: C, 55.10; H, 3.85; N, 10.71; found: C, 54.59; H, 3.94; N, 10.62.

**dichloro[picolinaldehyde O-methyl oxime]iron(II) Fe4**: Rufous solid, 275 mg, 87% yield. ATR-IR (cm^−1^): 2972, 1609, 1588, 1476, 1461, 1154, 1050, 946, 866, 785, 776. TOF-MS-ES+ (*m*/*z*): calcd. for [C_14_H_16_ClFeN_4_O_2_]^+^: 363.0311, found: 363.0292. Anal.: calcd. For C_14_H_16_Cl_2_FeN_4_O_2_: C, 42.14; H, 4.04; N, 14.04; found: C, 39.76; H, 3.84; N, 13.32.

### 2.4. Polymerization Procedure

**Binary component-catalyzed polymerization**: The iron complex was weighed in a glove box and transferred to a dried Schlenk tube (25 mL). Under argon, the quantitative solvent, isoprene, and the aluminum reagent were added. At the end of the reaction, HCl solution in methanol (methanol/HCl = 50/1) was added to quench the polymerization, and the mixture was poured into a large volume of methanol with 2,6-bis(1,1-dimethylethyl)-4-methylphenol as a stabilizing agent. The solid was collected and washed several times with methanol before being dried under vacuum at 50 °C for 15 h. The polymer yield was determined by gravimetry.

**Ternary component-catalyzed polymerization**: The iron complex was weighed in glove box and transferred to a dried Schlenk tube (25 mL). Under argon, the quantitative solvent and isoprene were added. Following that, a solution of [Ph_3_C][B(C_6_F_5_)_4_] in toluene (1.0 equiv. to iron catalyst) and cocatalyst were added. At the end of the reaction, HCl solution in methanol (methanol/HCl = 50/1) was added to quench the polymerization, and the mixture was poured into a large volume of methanol containing 2,6-bis(1,1-dimethylethyl)-4-methylphenol as a stabilizing agent. The solid was collected and washed several times with methanol before being dried under vacuum at 50 °C for 15 h. The polymer yield was determined by gravimetry.

## 3. Results and Discussion

### 3.1. Synthesis and Characterization of Ligands ***L2***, ***L4*** and Fe(II) Complexes

Following the reported procedure, ligands **L2** and **L4** were prepared by Schiff base condensation and were characterized by NMR and IR. Ligands **L1** and **L3** were used as received. After that, the former ligands were treated with FeCl_2_ in methylene dichloride at room temperature to afford various of pyridine-oxime-ligated iron(II) complexes which were characterized by IR, high resolution mass spectroscopies and elemental analysis (Figure 1). The IR spectra showed a shift in the characteristic *v*_(C=N)_ bands (1650–1604 cm^−1^) of the iron complexes toward higher wave numbers than the *v*_(C=N)_ bands (1586–1593 cm^−1^) observed for the corresponding free ligands, an indication of the coordination of ligands to the iron center. Krause et al. [47] postulated that transition metal complexes bearing pyridine-2-aldoxime and containing unionized oxime proton have *v*_(CH=N)_ bands in the range of 1654–1614 cm^−1^ and *v*_(N-O)_ bands in the range of 1069–1036 cm^−1^, respectively. Furthermore, the *v*_(OH)_ stretching frequency of metal complexes with unionized oxime proton displayed multiple bands between 3194 and 2791 cm^−1^, instead of the broad band at 3250 cm^−1^ of *v*_(OH)_ observed with free ligands [48]. Therefore, similar to Krause’s findings, these IR data for present iron complexes (**Fe1**–**Fe3**) showed that oxime protons are unionized (Appendix A).

In order to demonstrate the bonding and coordinating mode, a monocrystal of **Fe2** that would be suitable for X-ray diffraction was grown by the slow diffusion of hexane into a dichloromethane solution at −30 °C in glovebox (The detailed data were displayed in Appendix A). The molecular structure with selected bond distances and angles is described in Figure 2 (CCDC number: 2160530). These data also demonstrate the existence of two unionized oxime protons. The structure of **Fe2** consists of two ligands and two chloride atoms and adopts a distorted octahedron geometry in which two N_oxime_ atoms occupy the apical position [N(2)^1^-Fe-N(2) = 175.2(2)] and N(1)_pyridine_, N(1)^1^_pyridine_, Cl(1) and Cl(1)^1^ atoms form the basal equatorial plane. Interestingly, the coordination environments of the two ligands were exactly same and the bond lengths of each Fe-N_pyridine_, Fe-N_oxime_ and Fe-Cl were identical [Fe-N(1) = Fe-N(1)^1^ = 2.283(5), Fe-N(2) = Fe(1)-N(2)^1^ = 2.159(4), Fe-Cl(1) = Fe-Cl(1)^1^ = 2.4871(16)]. It is worth noting that the protons on O_oxime_ atoms point toward the chloride atoms, indicating the presence of hydrogen bonds; these results were similar to those for Ni complexes bearing pyridine-oxime ligands, as reported by Mukherjee [42]. The bond lengths of N_oxime_-O [1.397(5) Å] are closer to those of a free oxime ligand (1.4 Å), and the bond angles of C=N_oxime_-O [115.3(4)°] match well with the reported complexes [47,48]. To further study the electronic properties of the iron center, the ^57^Fe Mössbauer spectrum of complex **Fe1** was collected at 90 K, as displayed in Appendix A. However, three quadrupole doublets were noticed for the solid samples, which clearly indicated that the sample was composed of high-spin low-spin states.

### 3.2. Isoprene Polymerization with Binary Catalytic System

As shown in Table 1, iron complexes **Fe1**–**Fe4** were investigated for isoprene polymerization combined with methylaluminoxane (MAO) in toluene. They showed medium-to-high activities at room temperature. In addition, the GPC analyses of polymers catalyzed by all iron complexes showed unimodal and narrow distributions (1.8–3.5), proving the occurrence of typical single site catalysis (Figure 3A). Complexes **Fe1**, **Fe3** and **Fe4** revealed relatively high activities (8.2 × 10^5^ g/mol·h) with about 50% of *cis*-1,4-selectivity, and high molecular weight polymers were obtained (Table 1, entries 1, 5, 6). However, complex **Fe2**, bearing one 6-methyl substituent at the pyridine group, showed lower activity because of steric hindrance. Notably, 98% conversion could be achieved in 5 h via catalysis with **Fe2** (Table 1, entry 4). Interestingly, the substituent at 6-position of pyridine also slightly affected the selectivity to produce some *trans*-1,4-polyisoprene; the *trans*-1,4-unit gradually increased with an increase in the conversion rate. We attributed the variation in *trans*-1,4-selectivity to the steric effect of the progressively larger polymer chains. Likewise, the PDI of polyisoprene obtained with complex **Fe2** increased as the reaction time lengthened, albeit with decreasing molecular weight, suggesting fast chain transfer due to the large amount of MAO that was present (Figure 3B).

Subsequently, the effects of the reaction parameters on catalytic performance, e.g., the amount of MAO and reaction temperature, were systematically studied using **Fe1** and **Fe4**. The results are summarized in Table 2. For precatalyst **Fe1**, the catalytic activities remained largely unchanged as the [Al]/[Fe] ratio decreased from 500 to 50, resulting in around a 91% yield after 10 min when the [Al]/[Fe] ratio was 50 (Table 2, entry 3). However, the catalytic activities went down by decreasing the [Al]/[Fe] amount to 10, whereby a 54% yield of polyisoprene in 10 min was obtained (Table 2, entry 4) and a 98% yield by prolonging the reaction time to 1 h (Table 2, entry 5). Fortunately, even if the amount of MAO was reduced to 0.05 mmol, polymerization could still be carried out with 92% yield in 120 min (Table 2, entry 7), indicating the excellent catalytic properties of complex **Fe1** with a low amount of cocatalyst. It was interesting to note that reducing the [Al]/[Fe] ratio from 500 to 10 led to a decrease in the polymer molecular weight, probably due to the stability of the catalyst transition state maintaining chain propagation rather than causing chain termination/transfer in the presence of excessive MAO (Figure 3C). Variation of the [Al]/[Fe] ratio did not have obvious effects on the molecular weight distribution or polymerization selectivity. Furthermore, in order to study the role of oxime hydroxyl groups, complex **Fe4** was synthesized and catalytic isoprene polymerization was conducted. The results showed that complex **Fe4** possessed higher catalytic activities than **Fe1**; this was ascribed to the electron-donating properties of the oxime methoxyl groups, which facilitated isoprene coordination to the Fe center. For example, when reactions were quenched after 1 min under the same conditions with an [Al]/[Fe] ratio of 100, complex **Fe4** (Table 2, entry 9) exhibited the highest activity, i.e., 6.5 × 10^6^, compared to precatalyst **Fe1**, with 4.8 × 10^6^ (Table 2, entry 2). These results may also indicate that there was an induction period of **Fe1** to destroy hydrogen bonds between oxime hydroxyl groups and chloride atoms. To continue the comparison, temperature dependence tests were conducted with complexes **Fe1** and **Fe4**. When the polymerization temperature was −30 °C, the catalyst systems of **Fe1** (Table 2, entry 13) and **Fe4** (Table 2, entry 15) maintained high levels of activity with only marginal decrease. For catalyst **Fe1**, the *cis*-1,4-unit was better-suited to the low temperature, i.e., from 47% at 25 °C to 62% at −30 °C. Meanwhile, a high temperature slightly elevated the 3,4-structure (Table 2, entry 14). However, the high-temperature resistance of complex **Fe4** was lower than that of the former, and the polymer yield decreased to 74% at 70 °C (Table 2, entry 16), indicating low thermal stability. Upon expanding the application of this new catalyst, polymerization was catalyzed by complex **Fe1** with [Fe]/[Ip]/[MAO] = 1/20000/500 to achieve an 85% yield of polyisoprene in 2 h (Table 2, entry 17).

### 3.3. Isoprene Polymerization with Ternary Catalytic System

The catalytic performance of a ternary catalytic system using **Fe1**–**Fe4**, together with [Ph_3_C][B(C_6_F_5_)_4_] and trialkylaluminium, is summarized in Table 3. The polymerization results showed that the substituent at the 6-position of pyridine group also significantly affected activity and selectivity of isoprene polymerization because of the crowded iron center. In the case of complex **Fe2**, when AlEt_3_ was used as a cocatalyst, no polyisoprene was produced (Table 3, entry 3), and a fairly low yield was obtained by activating with Al(*i*-Bu)_3_ (Table 3, entry 4). Free 6-substitute catalysts (**Fe1**, **Fe3** and **Fe4**) showed higher activity than complex **Fe2** or previously reported imine ligated iron catalysts [31,35], regardless of cocatalyst type. Interestingly, when the Ph-substituent was added at the C=N position, complex **Fe3** showed the highest activity, i.e., up to 4.0 × 10^6^ g/mol·h (Table 3, entry 5). This may have been due to the steric effect of phenyl accelerating dealkylation by [Ph_3_C][B(C_6_F_5_)_4_]. In addition, it was clearly seen that the pyridine-oxime-ligated **Fe1** had big advantages in terms of these activities over complex **Fe4**, further emphasizing the particularity of oxime hydroxyl group. Otherwise, the ternary catalytic system provided polyisoprene with higher PDI than the binary one, but possessed similar selectivity and molecular weight. The results of previous works on iron-catalyzed isoprene polymerization are compared with those presented in this work in Appendix A. Notably, the pyridine-oxime-ligated iron complexes showed higher activities in a ternary catalytic system.

## 4. Conclusions

Iron complexes chelated with bidentate pyridine-oxime ligands were developed and examined for use in isoprene polymerization. For the binary catalytic system activated by MAO, the iron catalysts exhibited activities up to 6.5 × 10^6^ g/mol·h, and produced polyisoprene with high molecular weight (0.6–6.5 × 10^5^ g/mol) and narrow PDI (1.7–3.5), even at very low temperatures. The catalytic performance of precatalysts **Fe1**–**Fe4** was further studied by combing with [Ph_3_C][B(C_6_F_5_)_4_] and trialkylaluminium, yielding polymers at up to 98% yield in 2 min and suggesting the high catalytic capacity of these compounds. Steric influences on the catalytic activity and selectivity of complex **Fe2** were also clearly represented both in the binary and ternary catalytic methods. Moreover, these Fe complexes demonstrated high thermal stability regarding both activity and molecular weight. In summary, the newly developed, unionized, pyridine-oxime-ligated iron complexes are very promising compounds for isoprene polymerization applications. The development of other metal complexes suitable for pyridine-oxime ligand-catalyzed olefin polymerization is ongoing.

## Data Availability

The data that supports the findings of this study are available in the Appendix A of this article.

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
