# Peer review of "Synthesis, Characterization and Catalytic Property Studies for Isoprene Polymerization of Iron Complexes Bearing Unionized Pyridine-Oxime Ligands"

_polymers, 2022, doi:10.3390/polym14173612_

Round 1

Reviewer 1 Report

Mengmeng Zhao et al. report Synthesis, characterization and catalytic property studies for 2 isoprene polymerization of iron complexes bearing unionized 3 pyridine-oxime ligands. The work has revealed the catalytic activity of a new set of iron complexes on isoprene polymerization. The topic is highly exciting and the study is very informative.  This work is of high interest to readers working in the field of polyisopropene. 

Thus, it requires minor revisions in order to meet the journal's requirements.

  1. The authors have previously worked on several other iron complexes as catalysts on isoprene polymerization. I strongly believe a clear comparison (perhaps a new table can be introduced: this work and old work) with this work would be a great help to the reader.
  2. I am confused with the word unionized in the title and also in the manuscript. What does it mean and what is the difference with previous iron complexes? Is it mean previous complexes are ionized. As far as I understand that iron complexes are not ionized in common solvents (even in water). The authors can explain it. 
  3. I think the authors can reduce the no. of references. They have extended a bit as they presented this work. 

Author Response

Dear Reviewer,

Thank you for sending us the comments on our manuscript entitled " Synthesis, characterization and catalytic property studies for isoprene polymerization of iron complexes bearing unionized pyridine-oxime ligands" (Manuscript ID: polymers-1840496). After receiving the comments, we made in-depth thinking. These comments made us realize that our work has still some deficiency to be improved and perfected, and we have made significant modification to the manuscript.

Best regards,

Qinggang Wang

Comments to the Author

Mengmeng Zhao et al. report Synthesis, characterization and catalytic property studies for isoprene polymerization of iron complexes bearing unionized pyridine-oxime ligands. The work has revealed the catalytic activity of a new set of iron complexes on isoprene polymerization. The topic is highly exciting and the study is very informative. This work is of high interest to readers working in the field of polyisopropene.

Thus, it requires minor revisions in order to meet the journal's requirements.

Author’s response: Thanks for your supportive comments. A point-by-point response to the comments is given as below.

(1) The authors have previously worked on several other iron complexes as catalysts on isoprene polymerization. I strongly believe a clear comparison (perhaps a new table can be introduced: this work and old work) with this work would be a great help to the reader.

Author’s response: We thank the reviewer for insightful comments. The comparison of parts of previous works and this work were displayed in Page 11 and Table S4.

(2) I am confused with the word unionized in the title and also in the manuscript. What does it mean and what is the difference with previous iron complexes? Is it mean previous complexes are ionized. As far as I understand that iron complexes are not ionized in common solvents (even in water). The authors can explain it.

Author’s response: The word “unionized” in the title and manuscript is for pyridine-oxime ligands (L1-L3), which the oxime hydroxyl is unionized. In general, the oxime function coordinates to metal ions in the following two ways: unionized and ionized oxime. And in this work, we have indicated that the iron complexes were ligated with unionized oxime by ATR-IR and X-ray single crystal diffraction rather than O-Fe bond. So, we described it as unionized ligand.

(3) I think the authors can reduce the no. of references. They have extended a bit as they presented this work.

Author’s response: We thank the reviewer for insightful comments. Some references with less relevant to isoprene polymerization have been removed.

Reviewer 2 Report

                In their manuscript entitled “Synthesis, characterization and catalytic property studies for isoprene polymerization of iron complexes bearing unionized pyridine-oxime ligands” by Wang and coworkers synthesized two iron complexes based on 2-pyridyl oxime ligands, characterized the complexes to a limited extent, and then screened the complexes for their ability to polymerize isoprene with various additives (MAO, trityl cation, trialkylalumnium). In general I believe the work is probably suitable for Polymers, however, I have some concerns about the manuscript regarding citations and characterization, which should first be addressed prior to acceptance.

1. The authors have synthesized four complexes, however, some of these complexes have been previously described in some limited detail, yet no citation is given. This should be corrected:

Fe1/Fe2:  Indian Journal of Chemistry, Section A:  Inorganic, Physical, Theoretical & Analytical (1977), 15A(8), 696-9

Fe3: Croatica Chemica Acta (1981), 54(2), 173-82

2. Complexes Fe1, Fe3, and Fe4 have not been characterized by x-ray crystallography as far as I can tell by the authors or from the previous publications that have reported on these compounds (at least regarding Fe1 and Fe3). However, considering that these were more reactive, it would be appropriate to include x-ray data for these complexes as well, and contrast the bonding between Fe1, 3 and 4 with Fe2. Is it simply access to the metal that curtails reactivity of Fe2 as suggested, or are there significant differences in the bond lengths as well?

3. X-ray data should be submitted to the Cambridge Structural Database so that it is openly available to other researchers (as is standard) and the CSD #(s) provided in the manuscript.

4. No NMR data is provided for the iron complexes. While they are paramagnetic, it is often possible to characterize these by proton NMR. The coupling information will of course be lost, but the broad peaks can still be informative based on chemical shift/integration, and can be seen in some cases by expanding the range (I’ve personally seen Fe(II) signals as high as 200 ppm) – I would encourage the authors to provide this information, or if the signals are too broadened, to show an example spectrum.

5. Previous reports on these iron complexes reported on the spin state. I think it would be worth considering how the spin state (for example, are these S = 2, or is there sufficient distortion to achieve S = 1) may impact the chemistry. If the authors need to determine the spin state and do not have more specialized equipment to do that, I would recommend the Evans NMR method can be effective (example: Journal of Chemical Education (1997), 74(7), 815-816).

Round 2

Reviewer 2 Report

                In their revised manuscript entitled “Synthesis, characterization and catalytic property studies for isoprene polymerization of iron complexes bearing unionized pyridine-oxime ligands”, Wang and coworkers have addressed my comments. I agree that it is unfortunate that the authors were not able to provide x-ray data for the remaining complexes, as that would really have provided nice insights into the various reactivity. I appreciate that the authors were able to collect the Mössbauer spectrum for Fe1, which is interesting if most of the iron is in the low spin state. This would suggest that there is spin crossover behavior considering that they could not collect NMR data at presumably room temperature. I would recommend the following to the authors:

1.       The Mössbauer spectrum should be added to the manuscript or SI, and briefly discussed.

2.       The spin state of Fe1 and Fe2 under the reaction conditions (in this case 25 °C) should be compared, if not by Mössbauer at least by a magnetic moment determination. I would hypothesize that the Me, in addition to possibly limiting access to the metal, may also lead to extended N–Fe distances, which will lead to more high spin iron. This data may be of further use in defining the difference in reactivity between Fe1 and Fe2.

Author Response

Dear Reviewer,

Thank you for sending us the comments again on our manuscript entitled " Synthesis, characterization and catalytic property studies for isoprene polymerization of iron complexes bearing unionized pyridine-oxime ligands" (Manuscript ID: polymers-1840496). A point-by-point response to the comments is given as below.

Comments to the Author

In their revised manuscript entitled “Synthesis, characterization and catalytic property studies for isoprene polymerization of iron complexes bearing unionized pyridine-oxime ligands”, Wang and coworkers have addressed my comments. I agree that it is unfortunate that the authors were not able to provide x-ray data for the remaining complexes, as that would really have provided nice insights into the various reactivity. I appreciate that the authors were able to collect the Mössbauer spectrum for Fe1, which is interesting if most of the iron is in the low spin state. This would suggest that there is spin crossover behavior considering that they could not collect NMR data at presumably room temperature. I would recommend the following to the authors:

(1) The Mössbauer spectrum should be added to the manuscript or SI, and briefly discussed.

Author’s response: We thank the reviewer for their comments and we have added the Mössbauer spectrum in SI (Figure S2) and briefly discussed it in the manuscript (Page 5, signed in blue).

(2) The spin state of Fe1 and Fe2 under the reaction conditions (in this case 25 °C) should be compared, if not by Mössbauer at least by a magnetic moment determination. I would hypothesize that the Me, in addition to possibly limiting access to the metal, may also lead to extended N–Fe distances, which will lead to more high spin iron. This data may be of further use in defining the difference in reactivity between Fe1 and Fe2.

Author’s response: We thank you for the insightful comment and sorry for not being able to provide required experimental evidence. Although we can't prove them with single crystals or any testing methods in this work, the Me substituent at 6-position of pyridine exactly affect the N-Fe distance according to previous reports in the literature and our previous work (Dalton Trans., 2019, 48, 7862; J. Polym. Sci., 2020, 58, 2708), which leading to different space environment around iron center and further defining the catalytic activity of iron complex, and we have cited the reference in the manuscript (ref. 37).

Round 3

Reviewer 2 Report

I don't see a good reason why the authors cannot use the Evan's NMR method to determine the spin state of their catalyst system. The Evan's method can be performed with an NMR tube with a flame sealed capillary inside - all equipment that they should access to, and the experiment takes no longer than a standard 1D proton experiment. However, as they seem completely resistant to performing this measurement, I have no further suggestions.